# Association of Serum Vitamin B6 with All-Cause and Cause-Specific Mortality in a Prospective Study

**DOI:** 10.3390/nu13092977

**Published:** 2021-08-27

**Authors:** Donghui Yang, Yan Liu, Yafeng Wang, Yudiyang Ma, Jianjun Bai, Chuanhua Yu

**Affiliations:** Department of Epidemiology and Biostatistics, School of Health Sciences, Wuhan University, Wuhan 430071, China; yangdh@whu.edu.cn (D.Y.); liuyan202066@whu.edu.cn (Y.L.); wonyhfon@whu.edu.cn (Y.W.); 2020283050092@whu.edu.cn (Y.M.); 2015302170014@whu.edu.cn (J.B.)

**Keywords:** serum, vitamin B6, pyridoxal 5′-phosphate (PLP), mortality, NHANES

## Abstract

There is little evidence regarding the association between serum vitamin B6 concentration and subsequent mortality. We aimed to evaluate the association of serum vitamin B6 concentration with all-cause, cardiovascular disease (CVD), and cancer mortality in the general population using data from the National Health and Nutrition Examination Survey (NHANES). Our study examined 12,190 adults participating in NHANES from 2005 to 2010 in the United States. The mortality status was linked to National Death Index (NDI) records up to 31 December 2015. Pyridoxal 5′-phosphate (PLP) is the biologically active form of vitamin B6. Vitamin B6 status was defined as deficient (PLP < 20 nmol/L), insufficient (PLP ≥ 20.0 and <30.0 nmol/L), and sufficient (PLP ≥ 30.0 nmol/L). We established Cox proportional-hazards models to estimate the associations of categorized vitamin B6 concentration and log-transformed PLP concentration with all-cause and cause-specific mortality by calculating hazard ratios (HRs) and 95% confidence intervals (95%CIs). In our study, serum vitamin B6 was sufficient in 70.6% of participants, while 12.8% of the subjects were deficient in vitamin B6. During follow-up, a total of 1244 deaths were recorded, including 294 cancer deaths and 235 CVD deaths. After multivariate adjustment in Cox regression, participants with higher serum vitamin B6 had a 15% (HR = 0.85, 95%CI = 0.77, 0.93) reduced risk of all-cause mortality and a 19% (HR = 0.81, 95%CI = 0.68, 0.98) reduced risk for CVD mortality for each unit increment in natural log-transformed PLP. A higher log-transformed PLP was not significantly associated with a lower risk for cancer mortality. Compared with sufficient vitamin B6, deficient (HR = 1.37, 95%CI = 1.17, 1.60) and insufficient (HR = 1.19, 95%CI = 1.02, 1.38) vitamin B6 level were significantly associated with a higher risk for all-cause mortality. There was no significant association for cause-specific mortality. Participants with higher levels of vitamin B6 had a lower risk for all-cause mortality. These findings suggest that maintaining a sufficient level of serum vitamin B6 may lower the all-cause mortality risk in the general population.

## 1. Introduction

Micronutrients, including minerals and vitamins, are important components of our bodies. However, about two billion people lack key micronutrients globally [1]. Micronutrient deficiency can result in slow growth, high anemia prevalence, increased infection rates, and even death [2]. The 2017 Global Nutrition Report showed that one million premature deaths attributable to micronutrient deficiency have occurred annually [1]. As one of the important micronutrients, the prevalence of vitamin B6 deficiency was about 11% according to the second national report of the Centers for Disease Control and Prevention (CDC) [3]. Vitamin B6 plays a role as an essential coenzyme that regulates catabolic and anabolic processes [4]. It is involved in many enzyme activities as a cofactor, including molecule synthesis, interconversion, and degradation [5]. Researchers have found that suboptimal levels of vitamin B6 were associated with increased risk of chronic diseases such as diabetes [6], cancer [7], and cardiovascular disease (CVD) [8].

There are also several epidemiological studies examining the association of vitamin B6 with all-cause and cause-specific mortality [9,10,11]. Results from cohort analyses including middle-aged and elderly individuals in Shanghai showed that a high dietary intake of vitamin B6 was inversely associated with all-cause and CVD mortality [9]. Cui et al. found that a high dietary intake of vitamin B6 was inversely associated with CVD mortality among Japanese subjects [10]. Another study demonstrated that compared with the lowest tertile of intake of vitamin B6, the highest tertile was associated with a lower risk of all-cause and cancer mortality [11]. Although many previous studies evaluated the associations of vitamin B6 intake with all-cause and cause-specific mortality, the association of serum vitamin B6 concentration with all-cause and cause-specific mortality has not attracted much attention.

Therefore, we used data from the National Health and Nutrition Examination Survey (NHANES) in this study and prospectively investigated the association of serum vitamin B6 with all-cause and cause-specific mortality in the general population of U.S. adults. Additionally, this study emphasized the importance of maintaining a sufficient level of vitamin B6 in our bodies.

## 2. Materials and Methods

### 2.1. Study Population

NHANES is a repeated nationally representative cross-sectional survey carried out by the National Center for Health Statistics (NCHS) of the CDC. Briefly, NHANES use a multistage probability sampling design to collect information on health and nutritional status in the United States. More details on survey design and methods in NHANES have been previously published elsewhere [12]. The NCHS Research Ethics Review Board has approved the NHANES, and each survey participant provided informed consent. We used information from this publicly available and deidentified NHANES database so that our study was exempt from review by the Institutional Review Board. The concentration of vitamin B6 was measured from 2003 to 2010 in four cycles of NHANES. However, due to the large difference between the method of measurement in 2003–2004 and that in 2005–2010 and no available adjustment to make them comparable, we only used NHANES data from three cycles, that is, from 2005 to 2010 in this study. Detailed information is displayed on the website of NHANES (www.cdc.gov/nchs/nhanes/index.htm; accessed on 8 April 2021). The concentration of serum vitamin B6 was measured from 2005 to 2010 in 21,281 NHANES participants. Of the 21,281 participants, we excluded 827 participants with extreme values (>Q3 + 3*IQR) of vitamin B6 concentration. Moreover, we excluded 4911 participants without information on mortality and survival status. Furthermore, we excluded 3353 participants missing data on marital status, educational level, family income, body mass index (BMI), smoking status, drinking status, physical activity level. Finally, our study included 12,190 participants.

### 2.2. Measurement of Vitamin B6

From 2005 to 2010, serum vitamin B6 concentration was measured in NHANES participants by reversed-phase high-performance liquid chromatography (HPLC) using fluorometric detection, in which post-column introduction of a sodium chlorite derivatization was incorporated to improve the signal. The detection limit of HPLC was 0.3 nmol/L. Samples were stored at −70 °C and were stable for at least 5 years. Refrigerated samples could be used if they were brought promptly from the site of collection within 2 h.

### 2.3. Ascertainment of Mortality

Ascertainment of all-cause and cause-specific mortality was performed through linkage to National Death Index (NDI) records until 31 December 2015. The cause of death was determined according to the 10th revision of the International Classification of Disease (ICD-10). The main outcomes in our study were all-cause, CVD, and cancer mortality. CVD mortality was defined as ICD-10 codes I00–I09, I11, I13, I20–I51, and I60–I69, and cancer mortality was defined as ICD-10 codes C00–C97.

### 2.4. Assessment of Covariates

Information on sociodemographic factors and lifestyle factors at baseline was obtained via examination and questionnaire-based interviews. Sociodemographic factors included age, sex, race/ethnicity, marital status, education level, family income. Lifestyle factors included BMI, consumption of cigarettes and alcohol, and physical activity level. Race/ethnicity was categorized as Hispanic, non-Hispanic white, non-Hispanic black, or other non-Hispanic. Education level was separated into three levels: less than high school; high school or equivalent; college or above. Marital status was classified into three levels: married or living with partner; widowed, divorced, or separated; never married. Family income-to-poverty ratio was divided into three levels: 0–1.0; 1.1–3.0; >3.0 BMI was equal to weight (kilogram) divided by height (meter) squared and categorized as normal weight or underweight (<25 kg/m^2^), pre-obese (25–29.9 kg/m^2^), obese (≥30 kg/m^2^). Drinking status was classified as never drinker, former drinker, or current drinker. Smoking status was separated as never smoker, former smoker, or current smoker. Physical activity was categorized as whether the participants met or not the recommendation to perform at least 150 min of moderate to vigorous physical activity (MVPA) (75 min of vigorous physical activity or 150 min of moderate physical activity) each week as reported in the 2020 World Health Organization (WHO) Physical Activity Guidelines [13].

### 2.5. Data Analysis

Due to the complex design of NHANES, sample weights, clustering, and stratification were taken into consideration in all analyses. Survival time was calculated from the date of measurement of serum vitamin B6 concentration to the time of death or the end of the study period (31 December 2015), whichever came first. Age was described as mean value with standard error (SE). Categorized variables were shown as frequency with percentage. One-way analysis of variance (ANOVA) with Bonferroni test was used to examine age differences between three groups, while the chi-square test was implemented to test the difference of categorized variables between three groups. Cox proportional hazards regression models were established to examine the association of serum vitamin B6 with all-cause and cause-specific mortality by calculating hazard ratios (HRs) and 95% confidence intervals (CIs). Pyridoxal 5′-phosphate (PLP) was the biologically active form of vitamin B6 and was used to reflect the level of vitamin B6 in our body [14,15]. According to the definition of vitamin B6 status, serum vitamin B6 was categorized as deficient (PLP < 20 nmol/L), insufficient (PLP ≥ 20.0 and <30.0 nmol/L), and sufficient (PLP ≥ 30.0 nmol/L) [16]. Serum PLP concentrations were log-transformed due to non-normal distribution and analyzed as the continuous variable. We used three models to investigate the association of serum vitamin B6 with all-cause and cause-specific mortality. In model 1, we only adjusted for age (<60 years; ≥60 years) and sex (male; female). In model 2, we further adjusted for race/ethnicity (Hispanic; non-Hispanic white; non-Hispanic black; other non-Hispanic), marital status (married or living with partner; widowed, divorced, or separated; never married), education level (less than high school; high school or equivalent; college or above), family income/poverty ratio (0–1.0; 1.1–3.0; >3.0). In model 3, we further adjusted for BMI (normal weight or underweight [<25 kg/m^2^]; pre-obese [25–29.9 kg/m^2^]; obese [≥30 kg/m^2^]), smoking status (never smoker; former smoker; current smoker), drinking status (never drinker; former drinker; current drinker), physical activity level (<150 min MVPA/week; ≥150 min MVPA/week). We also calculated the E-value, which was the minimum strength of association of unmeasured confounders with treatment and outcome on the risk ratio scale [17]. In addition, we explored potential nonlinear associations between log-transformed PLP concentrations and mortality using restricted cubic spline regression with four knots (5th, 35th, 65th, 95th) after multivariate adjustment as mentioned above.

To test the robustness of the results, we conducted several sensitivity analyses in our study. First, we examined the associations after excluding participants who died in the first two years of follow-up to reduce the potential role of reverse causation. Second, as renal dysfunction may influence circulating vitamin B6 levels and cardiovascular events, we estimated renal function by calculating glomerular filtration rate (eGFR) according to the CKD-EPI Creatinine Equation (2009) [18] and further adjusted for renal function (eGFR ≥ 60 mL/min·1.73 m^2^; eGFR < 60 mL/min·1.73 m^2^). Third, repeated analyses were performed according to tertiles of serum vitamin B6. All data analyses were conducted using Stata version 15 (Stata Corp, College Station, TX, USA). Two-sided *p* < 0.05 was considered statistically significant.

## 3. Results

Of the 12,190 participants (6010 men [49.0%]; mean age [SE], 46.6 [0.4] years), 1244 died, including 294 cancer deaths and 235 CVD deaths. Table 1 shows the characteristics of the study population at baseline. Overall, the participants in our study were more likely to be women, Non-Hispanic White, and married. They tended to consume alcohol, never smoke, and met the recommendation of 150 min/week of MVPA. Most participants had high family income, educational level, and BMI. Vitamin B6 was sufficient in the majority of participants (70.6%), and only 12.8% of the participants were deficient in vitamin B6. Participants with higher PLP levels, in general, were more likely to be younger, married, non-Hispanic White, current drinkers, and less likely to smoke cigarettes and be obese and they tended to have higher education level and family income and perform more than 150 min MVPA each week.

After multivariate adjustment for sociodemographic factors and lifestyle factors in a cox proportional hazards model, deficient and insufficient vitamin B6 were significantly associated with a higher risk for all-cause mortality. There was no significant association of serum vitamin B6 level with cancer and CVD mortality. Compared with those with sufficient serum vitamin B6, participants with deficient serum vitamin B6 had a 37% (HR = 1.37, 95%CI = 1.17, 1.60) higher risk for all-cause mortality. A high risk for all-cause mortality was also observed among those insufficient in vitamin B6 (HR = 1.19, 95%CI = 1.02, 1.38). The magnitude of unmeasured confounding needed to explain away these associations was 2.08 for deficient vitamin B6 and 1.67 for insufficient vitamin B6. For the association with cancer mortality, HR was 1.22 (95%CI = 0.85, 1.74) for deficient vitamin B6 and 1.38 (95%CI = 0.95, 2.02) for insufficient vitamin B6. Participants deficient and insufficient in vitamin B6 had HRs of at least 1.74 and 2.10 beyond the measured confounders, respectively. For the association with CVD mortality, the participants deficient and insufficient in vitamin B6 had a 26% (HR = 1.26, 95%CI = 0.89, 1.77) and a 7% (HR = 1.07, 95%CI = 0.65, 1.78) increased risk, respectively. The corresponding E-values were 1.83 for deficient vitamin B6 and 1.34 for insufficient vitamin B6. Our results also indicated a significant association of log-transformed PLP concentration with all-cause and CVD mortality. A per unit increment in the log-transformed PLP concentration was associated with a 15% (HR = 0.85, 95%CI = 0.77, 0.93) reduced risk for all-cause mortality and a 19% (HR = 0.81, 95%CI = 0.68, 0.98) reduced risk for CVD mortality. No significant association was observed between PLP concentration and CVD mortality (HR = 0.81, 95%CI = 0.66, 1.01). The association of serum vitamin B6 with all-cause and cause-specific mortality and E-value is shown in Table 2 and Table 3, respectively. The detailed association of serum vitamin B6 with all-cause and cause-specific mortality is shown in Appendix A.

In further analysis, restricted cubic spline regression with four knots (5th, 35th, 65th, 95th) after multivariate adjustment as mentioned above demonstrated a non-linear relationship between serum PLP and all-cause mortality (*p* for non-linearity = 0.03). Figure 1 shows the HRs for all-cause mortality in restricted cubic spline regression with three knots. The HR for all-cause mortality decreased steadily as PLP concentration increased and almost did not change when log-transformed PLP concentration was >4. A per unit increment in log-transformed PLP could reduce by 31% (HR = 0.69, 95%CI = 0.58, 0.83) the risk for all-cause mortality among participants with log-transformed PLP ≤ 4, while for log-transformed PLP concentration >4, the HR for the association of log-transformed PLP with all-cause mortality was 1.02 (95%CI = 0.80, 1.31). The HRs for cause-specific mortality are shown in Appendix A.

In sensitivity analyses, the majority of the results were consistent with our primary results. When excluding deaths within 2 years of follow-up, an association of deficient and insufficient vitamin B6 with all-cause mortality was observed but was not statistically significant (Appendix A). After further adjustment for eGFR, consistent results were obtained (Appendix A). Furthermore, when serum vitamin B6 was classified as tertiles, the participants with lower serum vitamin B6 had lower all-cause and cancer mortality (Appendix A).

## 4. Discussion

In the analysis of this nationally representative prospective cohort study of U.S. adults, we found that deficient and insufficient serum vitamin B6 were significantly associated with a higher risk for all-cause mortality. Participants deficient and insufficient in serum vitamin B6 had a higher risk for cancer mortality and CVD mortality, but these associations were not statistically significant. We also found a significant association of PLP as a continuous variable with all-cause and CVD mortality.

The prevalence of deficiency in serum vitamin B6 was 12.8% in our study, which was almost the same as that in a previous study (11%) [3]. This may be partly due to the fact that many common foods, such as various meats and vegetables, are rich in vitamin B6 and thus contribute to maintaining a sufficient vitamin B6 status [15]. To our knowledge, only a few studies investigated the association of serum vitamin B6 concentration with all-cause and cause-specific mortality [19,20,21,22,23]. A prospective cohort study examined the association of serum vitamin B6 concentration with all-cause mortality and found that vitamin B6 concentration, whether as a continuous variable or categorized variable, was significantly associated with all-cause mortality [19]. Meanwhile, Huang et.al. investigated the effects of the vitamin B group on all-cause mortality in elderly individuals in Taiwan and found that participants with sufficient vitamin B6 had a lower risk for all-cause mortality, which is almost consistent with our results [21]. Moreover, a prospective case–cohort Study also found that renal cell carcinoma patients with high PLP had a significantly lower risk for all-cause mortality, which supports our results [22].

In addition, a previous epidemiological study showed a significant association between serum continuous vitamin B6 and CVD mortality and an association of deficient vitamin B6 with CVD mortality [19]. Our study also found that vitamin B6, as a continuous variable, was significantly associated with CVD mortality. A research demonstrated that vitamin B6 is linked to cardiovascular events [24]. Vitamin B6 could reduce the concentration of serum homocysteine [25], which is a risk factor for cardiovascular disease [26]. As categorized variables, deficient and insufficient vitamin B6 corresponded to a higher risk for CVD mortality in our study, but the association was not significant for both vitaminB6 status. This may because that epidemiological study was conducted among renal transplant recipients. It was estimated that low circulating concentrations of PLP were common in renal disease patients [27]. A low PLP concentration and renal disease could both increase the risk for CVD [8,28]. Another study found that renal transplant recipients deficient in vitamin B6 had a worse functional vitamin B6 status than healthy people and renal transplant recipients sufficient in vitamin B6 [20]. Compared with the low vitamin B6 concentration common in renal disease patients, vitamin B6 level varied widely, and the majority of participants had a high concentration in our study. Consequently, although a low level of vitamin B6 could increase the risk for CVD mortality, a significant association of deficient vitamin B6 level with CVD mortality was more likely to be observed among renal disease patients than among the general population in our study.

In terms of cancer mortality, David et.al. found that PLP concentration was significantly associated with renal cell carcinoma mortality [22]. However, we did not observe a significant association between PLP level and cancer mortality. It was reported that vitamin B6 level is related to cancer events [29]. Vitamin B6 deficiency could lead to changes affecting genes, such as chromosome breaks and alterations in gene expression [30], which would increase the risk of cancer occurrence. The non-significant association in our study may be partly because the PLP concentration was lower in David’s study than in ours. In that study, only less than 50% of participants had sufficient vitamin B6 concentration, and 30% of them would be considered deficient in vitamin B6. However, in our study, 70.6% of participants had sufficient vitamin B6, and only 12.8% of them were deficient in vitamin B6.

In further analysis, regression with restricted cubic spline showed that a high PLP concentration was inversely associated with all-cause mortality and provided no more additional benefits. Another study reported similar results, indicating that the HR for all-cause mortality decreased with a moderate increase of PLP concentration, and the risk for all-cause mortality did not decrease with a high level of PLP [23]. Generally, adults in the United States take vitamin B6-containing supplements regularly [31]. High PLP concentrations would be observed shortly after taking vitamin B6-containing supplements, which would lead to a right-skewed distribution of vitamin B6 concentrations. Consequently, the risk associations at higher concentrations of vitamin B6 could be expected to tend toward the mean risk.

After excluding participants who died within 2 years of follow-up, it was observed that deficient and insufficient vitamin B6 levels were associated with an increased risk for mortality, but this association was not statistically significant. Of those 241 participants who died in the first 2 years of follow-up, 32.3% and 17.1% had deficient and insufficient vitamin B6, respectively. Moreover, 109 adult participants died from cancer and CVD. Compared with the proportion of deficient (12.8%) and insufficient (16.6%) vitamin B6, the proportion of participants deficient for vitamin B6 among those 241 adults was larger. This could reduce the HRs for mortality after excluding participants who died within 2 years of follow-up, especially for the association of deficient vitamin B6, and even lead to non-significant associations of deficient and insufficient vitamin B6 with all-cause mortality. Additionally, it is suggested that upcoming death or diseases may lead to reduced vitamin B6 concentration. Future studies can investigate the prospective effect of disease on vitamin B6 concentration.

This study has several strengths. First, we used data from a large national database, which is representative of the general population. Second, we investigated associations considering PLP both as a categorized variable and as a continuous variable. Third, we adjusted for many sociodemographic factors and lifestyle factors to estimate the association of serum vitamin B6 with all-cause and cause-specific mortality. However, there are still a few limitations in this study. First, our study has the common limitations of observational studies. Therefore, a random control trial is needed to explore the examined associations in the future. Second, we did not investigate the association of vitamin B6 with mortality in groups stratified by covariates. Future studies could evaluate this association in subgroups stratified by covariates and examine whether interactions exist. Third, considering the limitations of PLP as a biomarker, the ratio 4-pyridoxic acid (PA)/(pyridoxal + PLP) (PAr) was proposed to reflect the vitamin B6 status [32,33]. The association of PAr with all-cause and cause-specific mortality could be investigated in further analyses.

## 5. Conclusions

This study revealed that as a categorized variable, vitamin B6 was inversely associated with all-cause mortality after multivariate adjustment, including sociodemographic factors and lifestyle factors. Meanwhile, a higher log-transformed PLP concentration was associated with a lower risk of all-cause and CVD mortality. Hence, to reduce the risk of mortality, it is essential to address vitamin B6 and increase the concentration of PLP.

## Figures and Tables

**Figure 1 nutrients-13-02977-f001:**
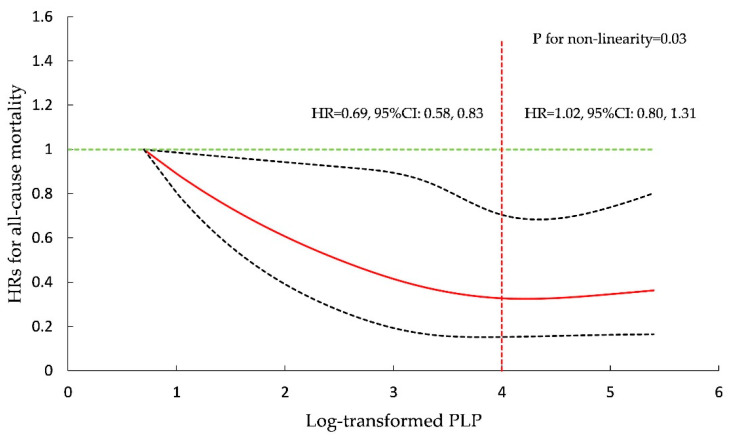
Association of pyridoxal 5′-phosphate (PLP) with all-cause mortality in restricted cubic regression with four knots (5th, 35th, 65th, 95th) in NHANES participants from 2005 to 2010.

**Table 1 nutrients-13-02977-t001:** Baseline characteristics of participants in NHANES from 2005 to 2010 ^a^.

	Total	Vitamin B6			*p* Value
		Deficient (12.8%)	Insufficient (16.6%)	Sufficient (70.6%)	
Age, mean (SE)	46.6 (0.4)	50.1 (0.6)	46.5 (0.5)	46.0 (0.4)	<0.01
Sex, %					<0.01
Male	49.0	35.3	39.5	53.8	
Female	51.0	64.8	60.6	46.2	
Race/ethnicity, %					<0.01
Hispanic	12.1	9.6	12.1	12.6	
Non-Hispanic White	71.8	70.3	69.3	72.6	
Non-Hispanic Black	10.7	16.8	13.4	9.0	
Non-Hispanic other	5.4	3.4	5.2	5.8	
Educational level, %					<0.01
Less than high school	18.5	27.3	20.5	16.3	
High school or equivalent	24.3	29.7	27.0	22.7	
College or above	57.2	43.0	52.4	60.9	
Marital Status, %					<0.01
Married or living with partner	64.8	59.2	62.4	66.5	
Widowed, divorced, or separated	18.8	26.8	21.2	16.8	
Never married	16.3	16.1	16.4	16.7	
Family income-poverty ratio, %					<0.01
0.0–1.0	13.2	21.4	15.7	11.1	
1.1–3.0	35.7	43.6	38.6	33.6	
>3.0	51.1	35.0	45.7	55.4	
BMI, kg/m^2^, %					<0.01
<25.0	30.7	24.2	27.4	32.7	
25.0–29.9	33.3	26.40	28.4	35.8	
≥30.0	36.0	49.4	44.2	31.5	
Drinking status, %					<0.01
Never drinker	10.8	14.5	11.9	9.9	
Ever drinker	16.5	26.0	17.6	14.5	
Current drinker	72.7	59.5	70.5	75.6	
Smoking Status, %					<0.01
Never smoker	52.2	38.4	49.4	55.4	
Ever smoker	24.7	22.1	22.6	25.7	
Current smoker	23.1	39.6	28.1	19.0	
Physical activity, %					<0.01
<150 min MVPA	46.8	61.3	50.6	43.2	
≥150 min MVPA	53.2	38.7	49.4	56.8	

Abbreviation: NHANES, National Health and Nutrition Examination Survey; SE, standard error; BMI, body mass index; MVPA, moderate to vigorous physical activity. ^a^ The NHANES used a complex design. Weight was taken into consideration. Continuous variables are shown as weighted mean and SE, while categorized variables are described as frequency and weighted percentage.

**Table 2 nutrients-13-02977-t002:** HRs (95% CIs) of all-cause and specific-cause mortality according to serum vitamin B6 concentrations.

	Vitamin B6 Status			Per Unit Increment in Log-Transformed PLP
	Deficient	Insufficient	Sufficient	
All-cause mortality				
Number of deaths/total	296/1886	244/2157	704/8147	1244/12,190
Model 1	1.83 (1.55,2.15)	1.37 (1.17,1.60)	1.00	0.72 (0.65,0.80)
Model 2	1.50 (1.28,1.75)	1.20 (1.04,1.40)	1.00	0.82 (0.74,0.90)
Model 3	1.37 (1.17,1.60)	1.19 (1.02,1.38)	1.00	0.85 (0.77,0.93)
Cancer mortality				
Number of deaths	67	66	161	294
Model 1	1.72 (1.19,2.49)	1.61 (1.11,2.32)	1.00	0.68 (0.54,0.85)
Model 2	1.43 (1.00,2.03)	1.43 (0.98,2.11)	1.00	0.76 (0.61,0.95)
Model 3	1.22 (0.85,1.74)	1.38 (0.95,2.02)	1.00	0.81 (0.66,1.01)
CVD mortality				
Number of deaths	47	45	143	235
Model 1	1.65 (1.18,2.31)	1.29 (0.78,2.12)	1.00	0.70 (0.58,0.83)
Model 2	1.31 (0.93,1.85)	1.07 (0.66,1.73)	1.00	0.80 (0.67,0.97)
Model 3	1.26 (0.89,1.77)	1.07 (0.65,1.78)	1.00	0.81 (0.68,0.98)

Model 1: adjusted for age (<60 years; ≥60 years) and sex (male; female). Model 2: further adjusted (from Model 1) for race/ethnicity (Hispanic; non-Hispanic white; non-Hispanic black; other non-Hispanic), marital status (married or living with partner; widowed, divorced, or separated; never married), education level (less than high school; high school or equivalent; college or above), family income/poverty ratio (0–1.0; 1.1–3.0; >3.0). Model 3: further adjusted (from Model 2) for BMI (normal weight or underweight [<25 kg/m^2^]; pre-obese [25–29.9 kg/m^2^]; obese [≥30 kg/m^2^]), smoking status (never smoker; former smoker; current smoker), drinking status (never drinker; former drinker; current drinker), physical activity level (<150 min MVPA/week; ≥150 min MVPA/week).

**Table 3 nutrients-13-02977-t003:** E-values for the point estimates.

	HR (95%CI)	E-Value
All-cause mortality		
Deficient	1.37 (1.17,1.60)	2.08
Insufficient	1.19 (1.02,1.38)	1.67
Sufficient	1.00	NA
Cancer mortality		
Deficient	1.22 (0.85,1.74)	1.74
Insufficient	1.38 (0.95,2.02)	2.10
Sufficient	1.00	NA
CVD mortality		
Deficient	1.26 (0.89,1.77)	1.83
Insufficient	1.07 (0.65,1.78)	1.34
Sufficient	1.00	NA

## Data Availability

Publicly available datasets were analyzed in this study. Data can be found here: www.cdc.gov/nchs/nhanes/index.htm (accessed on 8 April 2021).

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
