# Peer review of "Association of Serum Vitamin B6 with All-Cause and Cause-Specific Mortality in a Prospective Study"

_nutrients, 2021, doi:10.3390/nu13092977_

Round 1

Reviewer 1 Report

It seems to me a very interesting work and with the number of patients analyzed the results can be very significant. A statistical analysis between the three groups, sufficient, insufficient and deficiency of each of the analyzed variables is only required, taking into account that there are three groups and therefore the Bonferoni modification will have to be presented, for example.
On the other hand, there are three models of multivariate analysis for the three causes of death analyzed, I think it would be convenient, although this complicates the presentation of the results, see each of the variables that are significant factors in tables corresponding to each model, with what would have a better vision of the contribution of each factor.
With these changes I think the article would improve in understanding and clarity.

Author Response

Dear prof., please see the attachment

Reviewer 2 Report

The authors have used publically available deidentified data from NHANES to assess the prospective relationship between serum PLP (a measure of vitamin B6 status) and future all-cause and cause specific mortality risk. The study is well described and the analyses performed are (mostly) sound.

Below are some minor points mostly regarding technical issues concerning the analyses and presentation.

Include summary statistics for length of follow-up.

Table 1. Avoid: superfluous/distracting elements e.g. the number count in each row. Keep: percentages (includes the essential and easy to read information). Include relevant statistical tests for difference between the three categories in a separate (fifth) column. Age: Is it mean (SE)? Either the mean (SD) or mean (range) should be given. 

Precision of estimates. Percentages with 2 decimals is unneccessary and probably not warranted by the data. Likewise, P-values: avoid too many digits (usually not warranted or supported by the data) e.g. P=0.001 not P=0.0096. P=0.7 not 0.7151 (as far as journal style allows).

Figure 1. X-axis: Log transformed data have no units. 4 on the x-axis, when back-transformed, corresponds to 54.6 nmol/L. Try making a figure with back-transformed tick-marks and round numbers like 20, 30, 50 etc. (Unfortunately, many software packages are limited in their capability of doing this). Y-axis: HR should (also) be plotted on a ratio scale i.e. 2 and 0.5 should be equidistant from 1. 

Figure 1. The drawback of using restricted cubic splines is the strong assumptions involved in choosing knot positions. Try penalised cubic splines with 4 four knots if you have the option. This uses an algorithm to automatically select the best "bending" points from the data. 

Sensitivity analysis: You report that the estimate for insufficient vitamin B6 was no longer significant after excluding the first 2 years of follow-up, but if you compare Table 2 and Table S1 the estimates for insufficient vitamin B6 barely changes, while that for deficient changes considerably. Thus there, is evidence to support some level or reverse causality for deficient vitamin B6. This should be discussed more accurately in the text. This is an example (alas too common) of overfocusing on the statistical significance dichotomy whereas the main interest should be placed on changes (or variation) in effect estimates. Also review how you discuss other results accordingly.

Language: I find the description of the study mostly adequate, however, I find it sligthly lacking regarding grammar, especially, the use of verb tense in some cases. 

Author Response

Dear prof., please see the attachment
